# Structural and biochemical basis for DNA and RNA catalysis by human Topoisomerase 3β

Xi Yang [1], Sourav Saha[1], Wei Yang [2], Keir C. Neuman [3] & Yves Pommier [1] ✉

In metazoans, topoisomerase 3β (TOP3B) regulates R-loop dynamics and mRNA translation, which are critical for genome stability, neurodevelopment and normal aging. As a Type IA topoisomerase, TOP3B acts by general acid-base catalysis to break and rejoin single-stranded DNA. Passage of a second DNA strand through the transient break permits dissipation of hypernegative DNA supercoiling and catenation/knotting. Additionally, hsTOP3B was recently demonstrated as the human RNA topoisomerase, required for normal neurodevelopment and proposed to be a potential anti-viral target upon RNA virus infection. Here we elucidate the biochemical mechanisms of human TOP3B. We delineate the roles of divalent metal ions, and of a conserved Lysine residue (K10) in the differential catalysis of DNA and RNA. We also demonstrate that three regulatory factors fine-tune the catalytic performance of TOP3B: the TOP3B C-terminal tail, its protein partner TDRD3, and the sequence of its DNA/RNA substrates.

All living organisms have at least one copy of a type IA DNA topoisomerase[1–3]. Type IA topoisomerases share a padlock-shaped topoisomerase-core at their N-terminus while their C-terminal tail size and structure is highly diversified across species[3,4]. They use a three-step strategy−DNA cleavage, strand passage, DNA rejoining−to alter DNA topology, including supercoil relaxation and decatenation[1,5–7]. Type IA topoisomerases employ general acid-base catalysis to cleave DNA. A conserved tyrosine residue acts as nucleophile to break the DNA phosphodiester backbone. To cleave DNA, the tyrosine forms a covalent bond with the 5′-cleaved DNA end (tyrosyl-phosphate), leaving a 3′-OH group at the other DNA end. A reverse phosphoryl transfer initiated by the nucleophilic 3′-OH rejoins the DNA[5,8,9]. A divalent metal ion is commonly observed at the enzyme reaction center, coordinated by three or four conserved acidic residues[10–12]. Employing divalent cations at the catalytic site to trigger a phosphodiesterase activity is a general feature of many nucleic acid enzymes[13–17]. For Type IA topoisomerase enzymes, divalent cations are required for DNA rejoining, whereas their participation in DNA cleavage remains controversial[12,14,18,19]. In comparison, Type II topoisomerases, which

share a comparable TOPRIM catalytic center with Type IA topoisomerases, are believed to use divalent cations for DNA cleavage[14,20]. The overall catalytic efficiency of both Type IA and Type II topoisomerases can be greatly enhanced by the addition of metal ions, typically $Mg^{2+}$ or $Mn^{2+}$ ions[21–23].

Although sharing a highly conserved topoisomerase core domain, Type IA topoisomerase subfamily members such as topoisomerase 1 (bacteria) and topoisomerase 3 employ slightly different amino acids for catalysis[9,12,14,24], and exhibit distinct kinetic features in DNA relaxation and decatenation[25–28]. Additionally, bacterial topoisomerases 1 and 3 enzymes have disparate DNA sequence selectivity for cleavage. Topoisomerase 1 favors a cytosine at the −4 nucleotide position (relative to the cleavage site) whereas topoisomerase 3 does not[22,29–32]. This difference in DNA-base preference reflects different amino acid residue compositions within their DNA binding grooves and varied DNA base recognition[9,24].

Human cells express two Type IA topoisomerases, TOP3A and TOP3B. Recent studies revealed that a subset of the Type IA topoisomerases including TOP3B, are both DNA and RNA

[1]Developmental Therapeutics Branch & Laboratory of Molecular Pharmacology, Center for Cancer Research, National Cancer Institute, NIH, Bethesda, MD 20892, USA. [2]Laboratory of Molecular Biology, National Institute of Diabetes and Digestive and Kidney Diseases, NIH, Bethesda, MD 20892, USA. [3]Laboratory of Single Molecule Biophysics, National Heart, Lung and Blood Institute, National Institutes of Health, Bethesda, MD, USA. ✉e-mail: pommier@nih.gov

topoisomerases[3,33–35]. TOP3B is a key player in regulating cellular mRNA metabolism, which is crucial for the translation of neurodevelopment-related genes[33,34,36,37]. TOP3B also plays a positive role in the replication of RNA viruses in the infected human cells, thus can be considered a potential anti-viral target[38]. Moreover, TOP3B was shown to suppress R-loop accumulation in multiple gene regions, thus facilitating transcription at these genomic loci[37,39]. Activities of TOP3B in cells are associated with a scaffold protein, TDRD3, that recruits TOP3B to relevant cellular targets[33,34,37]. This recruitment involves a strong interaction between Domain II of TOP3B and the OB-fold of TDRD3[11]. Binding to TDRD3 also improves the DNA relaxation catalytic activity of TOP3B in Drosophila[40].

So far, little is known regarding the basic mechanism of TOP3B-mediated RNA catalysis. It also remains unclear what are the general structural and catalytic features that make some Type IA topoisomerases better RNA topoisomerases than others. To address these questions, we carried out biochemical and single-molecule measurements of hsTOP3B using a collection of mutants and nucleic acid substrates to understand the differential enzyme catalysis of DNA and RNA. We describe the disparate features of DNA and RNA catalysis of TOP3B and characterize the roles of divalent metal ions and of a critical catalytic lysine residue in the DNA/RNA cleavage and rejoining reactions. We describe how the DNA and RNA topoisomerase activities are stimulated by the C-terminal domains of TOP3B and by TDRD3, resulting in enhanced enzyme processivity and increased catalytic rate. We also demonstrate that the sequence of DNA and RNA substrates can significantly influence the substrate association and catalytic behavior of TOP3B.

## Results

### DNA and RNA cleavage by TOP3B
A divalent metal ion is commonly observed at the active site of Type IA topoisomerase enzymes, including TOP3B (Suppl Fig. 1), and is essential for the catalytic efficiency of these enzymes[9,12,19,21]. We conducted TOP3B cleavage assays (Fig. 1a) at various concentrations of $Mg^{2+}$ and $Mn^{2+}$ (Fig. 1b, c) to examine the effect of metal ions on TOP3B's DNA and RNA cleavage activities. We used a 40-nt ssDNA oligonucleotide and its RNA counterpart as substrates, the sequence of which is derived from a segment of the DDX5 gene containing a proposed TOP3B in vivo target site[37]. We determined the DNA cleavage sites of TOP3B with sequence-specific DNA oligonucleotides as markers, and RNA cleavage sites were mapped with RNA alkaline ladders and RNase T1 ladders (Suppl Fig. 2). We observed that TOP3B can cleave DNA and RNA without addition of divalent ions (Fig. 1b, c), alike bacterial topoisomerase 1 enzymes[12,41]. Metal-ion titration curves indicated that introduction of moderate amounts (submillimolar) of $Mg^{2+}$/$Mn^{2+}$ stimulates the accumulation of cleavage products at steady state. The forward reaction of TOP3B was considerably enhanced by $Mn^{2+}$ addition (Fig. 1d). Without metal addition, the slow DNA cleavage can possibly be contributed to trace amounts of metal ions remaining in the reaction system, as basal cleavage activity was effectively suppressed by increasing the EDTA concentration (Suppl Fig. 3). In addition, we found that $Mn^{2+}$ stimulates the DNA binding of TOP3B (Fig. 1e). Consistent with the importance of metal ion binding in the catalytic site of TOP3B, mutating the catalytic glutamic acid residue E9 (as part of the conserved metal ion binding motif) to glutamine (E9Q) abolished both DNA and RNA cleavage in the presence of $Mn^{2+}$ (Suppl. Fig. 4).

### DNA but not RNA rejoining by TOP3B requires the addition of a divalent metal ion
The fundamental characteristic distinguishing topoisomerases from nucleases is their ability to rejoin the nucleic acid backbone without the assistance of DNA repair, which allows them to perform their catalytic reactions with high efficiency and without changing the primary nucleic acid sequence[5]. To examine the role of metal ions in DNA rejoining, we conducted TOP3B reversal assays by adding high salt (0.5 M NaCl) to the steady-state mixture of TOP3B and DNA, blocking the forward reaction but not reversal (Fig. 1a). Coupling high-salt addition with adjustment of metal ions to various concentrations allowed us to compare the impact of divalent cations on DNA rejoining (Fig. 1f, g). We found that TOP3B only rejoins DNA after adding metal ions in solution, and that the reversal rate is increased with elevated metal-ion concentrations. Compared with $Mg^{2+}$, $Mn^{2+}$ is more potent. The divalent cation-accelerated DNA reversal, together with an interrupted TOP3B-substrate association at a high metal concentration (Suppl Fig. 5), can explain the suppression of DNA/RNA steady-state cleavage at higher metal concentrations (Fig. 1b, c). The minimal amount of metal ions in the system enables DNA cleavage but cannot trigger DNA rejoining, possibly reflecting the varied metal-binding affinities of TOP3B in the pre-cleavage versus pre-rejoining state. The change of binding affinity is likely a result of an active-site rearrangement by the enzyme between two consecutive phosphoryl-transfer reactions, as suggested for bacterial topoisomerase 1[12] and observed with other related nucleic acid enzymes[15,42].

In contrast to DNA, cleaved RNA was readily rejoined by TOP3B without addition of metal ion (Fig. 1h). Yet, 1 mM EDTA annihilated RNA rejoining, suggesting that reversal of RNA cleavage may also require trace amount of divalent ion as cofactor.

### Lysine 10 (K10) of TOP3B promotes DNA cleavage and is required for DNA rejoining
Crystal structure of EcTOP3 in a complex with ssDNA shows that a lysine residue (K8) coordinates the DNA scissile phosphate group[24] (Fig. 2a), and thus may play a role in aligning the DNA substrate for phosphoryl transfer during DNA cleavage[14,19,24]. This lysine residue is conserved among topoisomerase 3 enzymes including TOP3B (Fig. 2b), while missing in the bacterial topoisomerase 1 and Type II topoisomerase families[9,14,24]. This observation suggested that K10 may act as a TOP3B catalytic residue.

To test this possibility, we replaced K10 of TOP3B with methionine. The mutant enzyme failed to cleave DNA in the absence of added divalent cation and only showed limited DNA cleavage upon addition of divalent cation (Fig. 2c). This result implies that removing the tertiary amine of K10 in TOP3B reduces the enzyme metal ion-binding affinity, thereby higher metal-ion concentrations are required to cleave DNA. The role of K10 is reminiscent of the K155 in *Mus musculus* Endonuclease V (MmEndoV), in which a side-chain amine coordinates the RNA scissile phosphate during RNA cleavage[17]. K155 substitution leads to inadequate scissile-phosphate alignment and inappropriate metal-ion association, interrupting the nuclease activity of MmEndoV[17]. Likewise, aligning the DNA scissile phosphate by K10 is also likely key for the binding of a divalent cation to TOP3B and DNA cleavage (Fig. 2e). The K10 mutation also abolished DNA rejoining by TOP3B even upon addition of divalent cations (Fig. 2d), suggesting an indispensable role of K10 alongside divalent cations in re-aligning the cleaved DNA ends for DNA rejoining (Fig. 2f). An efficient DNA scissile-phosphate alignment likely requires both K10 and R338 (Fig. 2e), as both conserved residues coordinate the DNA scissile phosphate in EcTOP3 (Fig. 2a, b). Moreover, replacement of the conserved arginine residue in EcTOP1 (R321) also led to higher divalent cation requirement for DNA cleavage and resulted in lack of DNA rejoining[43].

The catalytic role of the K10 of TOP3B was tested in cells. Expression of WT TOP3B and mutant TOP3B-K10M in human HEK293 cells[35] produced significantly lower nucleic acid cleavage complexes with the mutant than the WT enzyme (Suppl Fig. 6), consistent with our

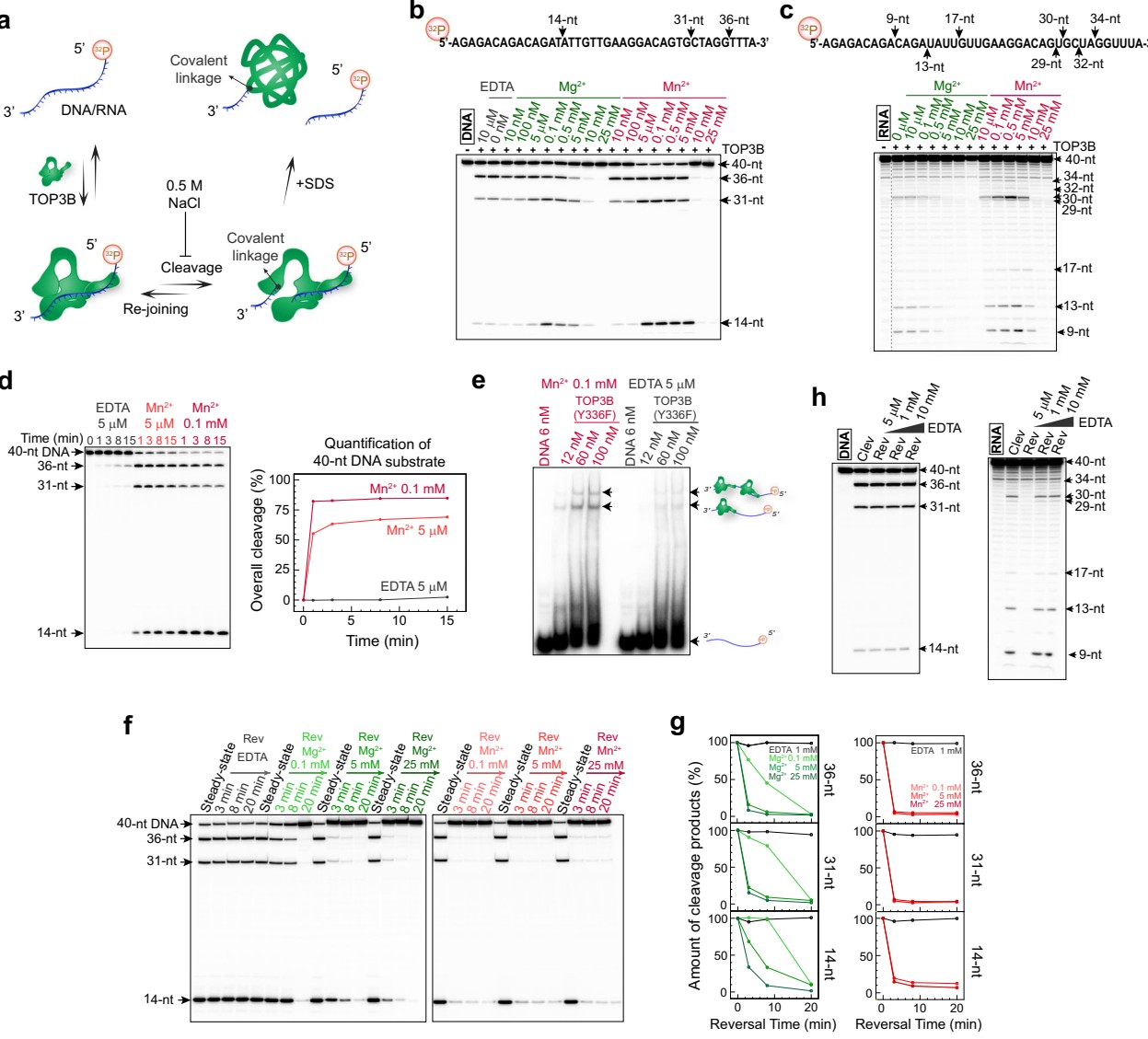

**Fig. 1 | TOP3B-mediated DNA and RNA cleavage and rejoining are differentially affected by Mg2+ and Mn2+. a** Schematic diagram showing DNA/RNA substrate labeling and TOP3B cleavage assay. **b, c** DNA and RNA cleavage by TOP3B at various Mg2+ and Mn2+ concentrations. Arrows indicate major DNA/RNA cleavage products and their corresponding cleavage sites in the substrate sequences. Gray dotted vertical line in **c** indicates combination of different lanes on the same gel with the same exposure condition. DNA cleavage sites were determined with sequence-specific DNA oligos as markers. RNA cleavage sites were mapped with RNA alkaline ladder and RNase T1 ladder (Suppl Fig. 2). **d** Time-course showing pre-steady-state DNA product formation in the absence of divalent cation (and with 5 µM EDTA) or with 5 µM or 0.1 mM Mn2+. Plot of DNA cleavage product versus time was obtained by analyzing the depletion of the 40-nt DNA substrate band. **e** EMSA assay with catalytic inactive mutant TOP3B-Y336F showing its DNA binding in the absence and

presence of Mn2+. TOP3B retardation bands are indicated by arrows and cartoons. **f** Time-course of DNA end-rejoining by TOP3B at various Mg2+ and Mn2+ concentrations. Steady state levels of cleavage were obtained by pre-incubating TOP3B with DNA for 30 min in solutions containing 0.1 mM Mg2+ or Mn2+. Subsequent reversal steps (Rev) were initiated by addition of 0.5 M NaCl and indicated amounts of divalent ions or EDTA. **g** Plots show reversal of DNA cleavage products over time (percentage) obtained by quantification of individual DNA products in panel **f**. **h** DNA and RNA cleavage-reversal assays at various EDTA concentrations. Cleavage (Clev) products were generated by incubating TOP3B with DNA/RNA substrate for 30 min in the presence of 5 µM EDTA, followed by reversal analysis (Rev) by adding 0.5 M NaCl and the indicated amounts of EDTA for 5 min. All the results of the gel-based assays in (**b–h**) are representative results obtained from 2–4 independent experimental repeats.

biochemical finding that K10 is critical for the DNA cleavage activity of TOP3B in vitro.

## Lysine 10 (K10) of TOP3B is implicated in RNA rejoining

RNA cleavage-rejoining experiments showed that TOP3B-K10M produces similar RNA cleavage levels as WT TOP3B in the absence of added divalent metal ions (Fig. 3a) and upon metal ion addition (Fig. 3b, c, lanes 2 vs. 6). However, TOP3B-K10M rejoined RNA breaks markedly slower than the WT enzyme (Fig. 3b). By removing both K10 and divalent metal ions (via EDTA chelation), RNA rejoining was completely blocked (Fig. 3c).

We therefore conclude that RNA cleavage can occur without K10 and metal-ion addition, whereas rejoining requires at least one of these two elements. This appears different for DNA processing where the DNA cleavage activity of TOP3B requires either K10 or divalent-metal addition, whereas rejoining requires both K10 and a metal addition. Therefore, compared to DNA, both cleavage and rejoining of RNA are less dependent on divalent ion addition and K10, indicating that RNA may be chemically and structurally a more favorable substrate than DNA for TOP3B catalysis. Namely, during RNA cleavage, the additional 2'-OH groups may make RNA strand more rigid and stable than DNA (e.g. via forming additional interactions with the surrounding amino

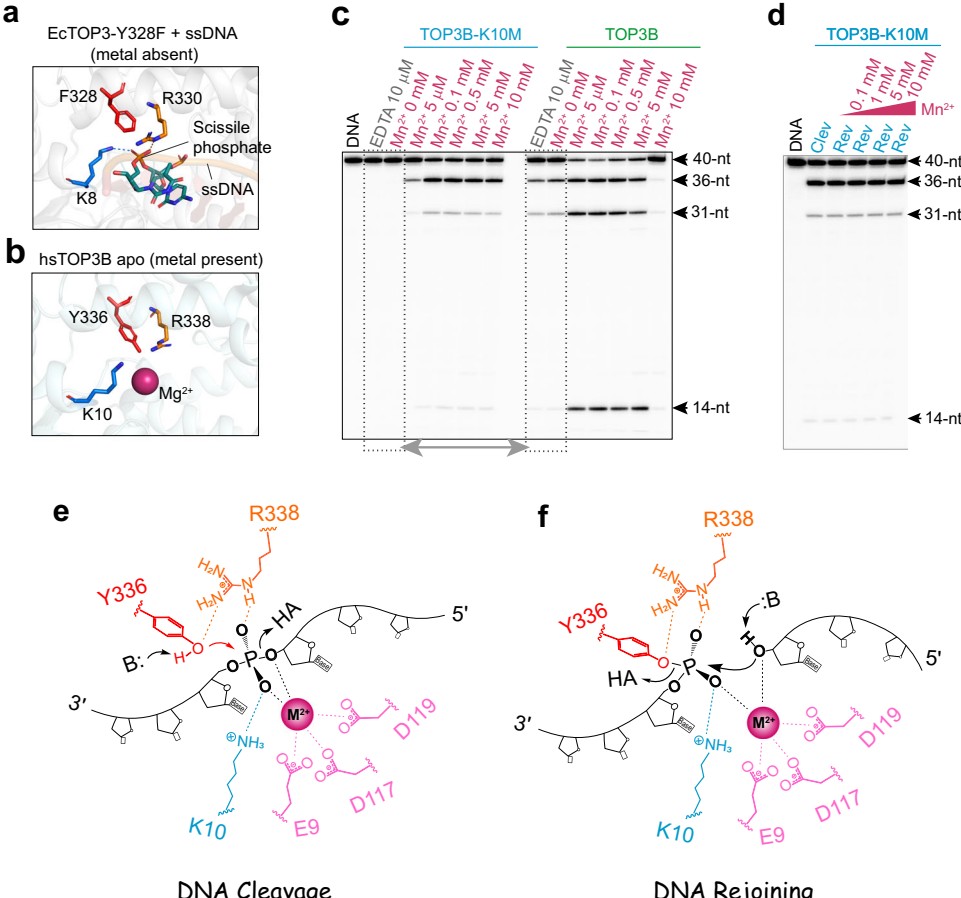

**Fig. 2 | TOP3B lysine 10 (K10) is a conserved catalytic residue assisting divalent cation-dependent DNA cleavage and required for DNA rejoining. a** Structure of *Ec*TOP3 (PDB ID: 1i7D) showing the interaction between the DNA scissile phosphate and the conserved Lysine residue K8 (blue dotted line) in the absence of divalent cation. Key residues for catalysis including the Tyrosine substitute F328 and Arginine R330 are shown as sticks. **b** Catalytic site of hsTOP3B apo structure (PDB ID: 5gvc) in the presence of a $Mg^{2+}$ ion showing the conserved Lysine K10 and other key catalytic residues. **c** DNA cleavage assay with WT TOP3B and mutant TOP3B-K10M without divalent cation and with the indicated concentrations of $Mn^{2+}$. Dotted boxes and double-headed arrow highlight the comparison of DNA cleavage activity between the mutant and the WT enzyme without divalent-metal addition. **d** DNA cleavage and lack of reversal with TOP3B-K10M in the presence of $Mn^{2+}$ (0.1–10 mM). DNA cleavage products were collected after 30 min incubation. Reversal assays were conducted by adding high salt and indicated amount of $Mn^{2+}$

for 5 min (Clev: Cleavage; Rev: Reversal). DNA cleavage assays in (**c**, **d**) were repeated twice independently with similar results. **e** Proposed mechanism for DNA cleavage by TOP3B. K10 and R338 of TOP3B coordinate two oxygen atoms in the DNA scissile phosphate, aligning the DNA substrate and facilitating the binding of a divalent cation (red-violet sphere). A properly bound metal ion, together with K10 and R338, stabilizes two non-bridging oxygen atoms and the 3′-oxygen leaving group (dotted lines), facilitating the nucleophilic attack by Y336 (red arrow) and subsequent phosphoryl transfer, resulting in a Tyrosyl phosphate group at the 5′-end of the cleaved DNA. **f** DNA-end re-alignment and rejoining. K10 and R338 play essential roles in re-aligning the 5′ DNA end (Tyrosyl phosphate) and facilitating divalent-metal binding. The captured metal ion further aligns the 3′-OH group as a nucleophile to trigger a reverse phosphoryl transfer to reform the DNA backbone. The potential general acid (HA) and base (B:) in (**e**, **f**) can be adjacent water molecules.

acid residues) thus enhancing the strand alignment for cleavage; and the 2′-OH group adjacent to the scissile phosphate can stabilize the 3′-O leaving group or act as a general acid to facilitate the process of phosphoryl transfer (Fig. 3d). Similarly, the 2′-OH can stimulate RNA rejoining by polarizing the nucleophile (the 3′-OH) or acting as a general base (Fig. 3e). This makes TOP3B catalytic activity resistant to either losing K10 or to an exceedingly low divalent ion concentration but not both. In addition, RNA is likely better than DNA in forming a catalytically competent complex with the enzyme, accelerating strand re-alignment necessary for rejoining.

## The C-terminal tail of TOP3B enhances the processivity and nucleic acid binding of TOP3B

Despite sharing a conserved topoisomerase core, type IA enzymes have diversified C-terminal tails[1,3,44]. These C-terminal domains with different sizes and sequences may contribute to the fine-tuning of enzymatic activities and/or cellular recruitment locations. For example, the C-terminal zinc-finger motifs of bacterial topoisomerase 1

enzymes stimulate DNA relaxation[44–46] and mediate a direct association between *Ec*TOP1 and RNA polymerase[47].

The C-terminal tail of TOP3B contains four conserved C4-type zinc-finger motifs as predicted by AlphaFold[48,49] and an Arg-Gly/Arg-Pro-rich segment near the C-terminus (Fig. 4a). By generating truncation mutants (Fig. 4b), we found that removal of the RG/RP-rich motif or the entire C-terminal domain of TOP3B results in a considerable reduction of DNA/RNA cleavage, while preserving the substrate sequence preference (Fig. 4c, d). This result indicates that the core domain of TOP3B is sufficient for catalytic activity, but that its activity is highly stimulated by the C-terminal tail. Gel shift experiments to measure DNA and RNA binding showed that this stimulation may result from enhanced DNA and RNA binding mediated by the C-terminal tail (Fig. 4e, f). We therefore propose that the Zinc-finger motifs and highly positively charged RG/RP-fragment in the C-terminal tail serve as nucleic acid binding modules that non-specifically bind DNA/RNA adjacent to the TOP3Bcore, increasing enzyme-substrate stability during catalysis (Fig. 4g).

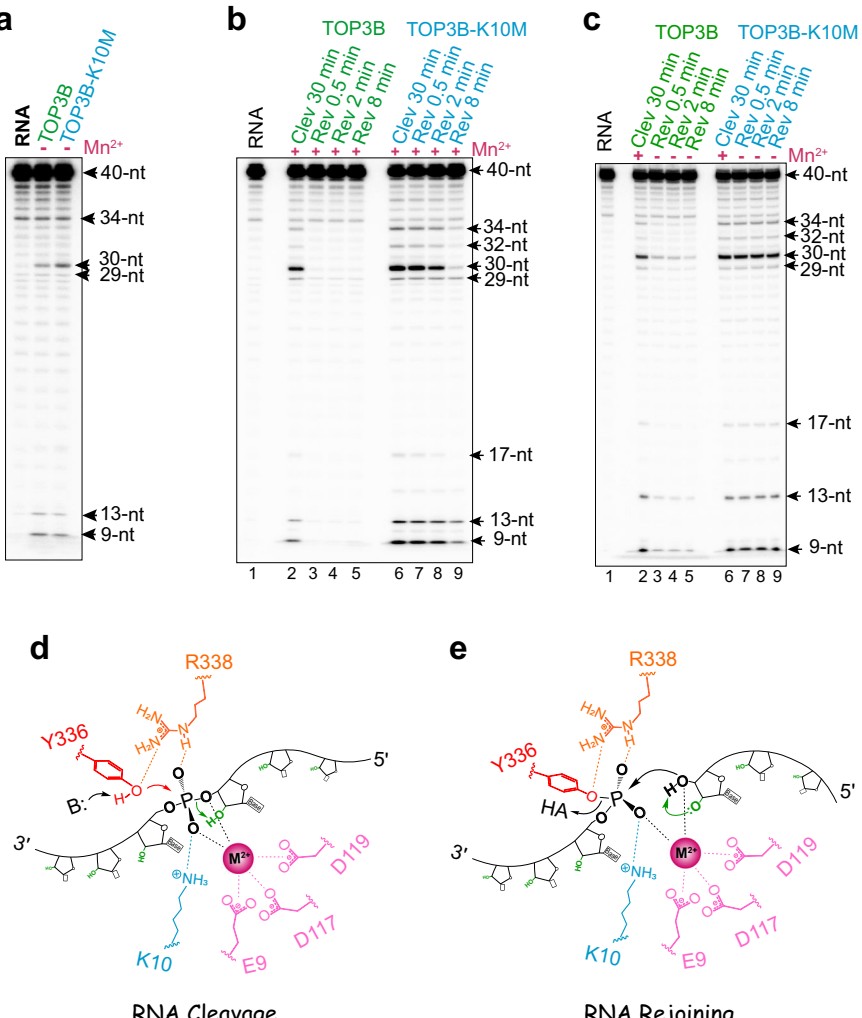

**Fig. 3 | TOP3B K10 is dispensable for RNA cleavage and rejoining. a** RNA cleavage by WT TOP3B and TOP3B-K10M without addition of divalent cations (and with 10 μM EDTA). **b** RNA cleavage and reversal in the presence of $Mn^{2+}$ (0.1 mM) showing a slower RNA rejoining by TOP3B-K10M compared to the WT enzyme. **c** Lack of RNA cleavage reversal by TOP3B K10 in the absence of added metal ion. Initial cleavage reactions for WT TOP3B and TOP3B-K10M (lanes 2 & 6) were conducted in the presence of 0.1 mM $Mn^{2+}$ and then followed by high-salt and EDTA addition (final EDTA concentration was adjusted to 50 μM for reversal). Each RNA assay in (**a**–**c**) was repeated twice independently with similar results. **d** Proposed mechanism for RNA cleavage by TOP3B is similar to DNA as described in Fig. 2e, except that the 2′-OH (colored in green) adjacent to the leaving group can readily stabilize the 3′-oxygen leaving group or serve as a general acid, promoting the process of phosphoryl transfer. **e** RNA end re-alignment and rejoining by TOP3B. The 2′-OH group (colored in green) can polarize the nucleophile (3′-OH) or act as a general base for RNA rejoining.

To determine how the C-terminal domain affects the turnover kinetics of TOP3B, we carried out single-molecule measurements[50–52] comparing DNA relaxation by TOP3B and the TOP3Bcore (Fig. 4h). Detailed descriptions of the experimental setup and procedures are provided in the methods section. In short, we inserted an 11-base DNA mismatch bubble into a 6-kb DNA tether to provide a ssDNA binding region allowing TOP3B to relax negatively supercoiled DNA. A typical TOP3B relaxation burst consists of discrete steps of DNA extension corresponding to catalytic cycles of a single enzyme (Fig. 4i). We obtained a step-amplitude distribution of TOP3B by converting the DNA extension change in each step (a single catalytic cycle) to changes in DNA linking number. As a representative type IA topoisomerase, we found that TOP3B removes one DNA supercoil (a linking number change of 1) per catalytic cycle (Fig. 4j). Compared to full-length TOP3B, which cycles multiple times in a single relaxation burst (within a single DNA binding event) thus removing multiple DNA supercoils, TOP3Bcore only removed a single DNA supercoil in a typical DNA relaxation burst, displaying extremely low processivity (Fig. 4k, l). Therefore, our data indicate that the C-terminal tail of

TOP3B converts the single-cycle TOP3Bcore into the processive molecular machinery of full-length TOP3B, by preventing the TOP3B-core from detaching from DNA (Fig. 4g).

### TDRD3 increases the catalytic processivity and rate of TOP3B

TDRD3 has been shown to recruit TOP3B to its sites of action in cells, guiding TOP3B to multiple cellular processes including transcription and translation[33,34,37]. In addition, Drosophila TDRD3 has been shown to stimulate the relaxation of hyper-negative supercoils by Drosophila TOP3B[40]. However, the molecular mechanism underlying this functional stimulation has not been studied.

We found that mixing purified human TOP3B and TDRD3 resulted in a stable heterodimer in solution (Suppl Fig. 7), which is consistent with the prior report that TDRD3 forms a stable interaction with domain II of the TOP3Bcore (Fig. 5a)[11]. Based on this structural information, we performed DNA and RNA cleavage-rejoining experiments at different molar ratios of TDRD3 and TOP3Bcore and full-length enzyme (Fig. 5). Addition of TDRD3 strongly activated the DNA and RNA cleavage activities of TOP3Bcore (Fig. 5b, c). Similarly, DNA and

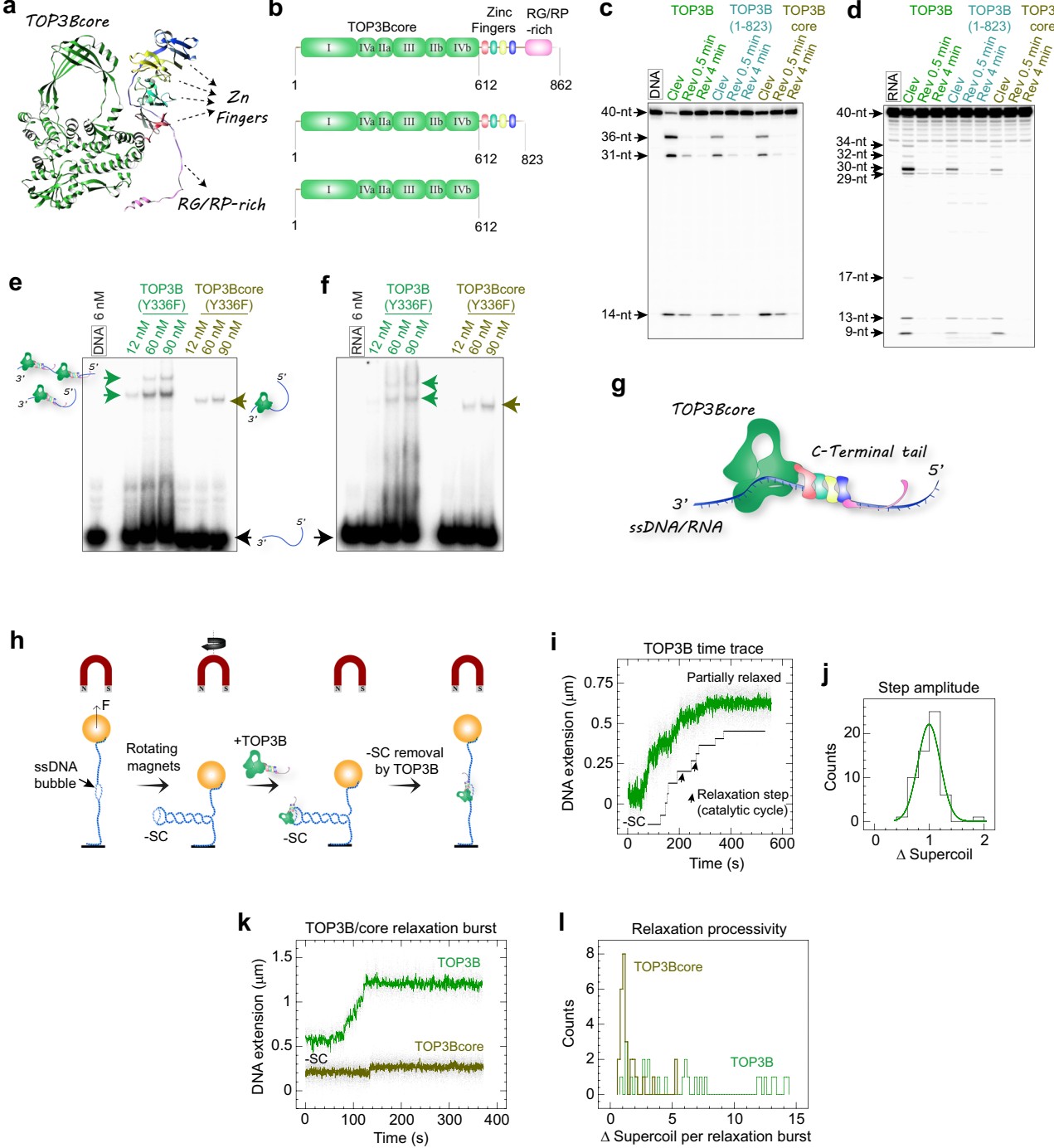

**Fig. 4 | The C-terminal tail of TOP3B stimulates DNA/RNA binding and improves the catalytic processivity of TOP3B. a** Overall architecture of TOP3B based on AlphaFold-predicted 3D model. The conserved TOP3Bcore is colored in green and the C4-type zinc-finger motifs and an Arg-Gly/Arg-Pro-rich fragment in the C-terminal tail are indicated by arrows. **b** Domain structures of the full-length TOP3B, of the RG/RP-rich fragment depleted mutant and of the TOP3Bcore. Roman numerals refer to the conserved domains I-IV of typical type IA topoisomerases and numbers to amino-acid positions. **c** DNA cleavage and reversal assays with WT TOP3B and mutants displayed in panel b. Same molar concentrations of DNA/RNA and enzymes were tested in all the assays. Steady-state cleavage products (Clev) and high-salt reversal products (Rev) at various time points are indicated above each lane. **d** same as c for RNA cleavage and reversal. **e**, **f** EMSA assays using catalytic inactive TOP3B-Y336F and TOP3Bcore-Y336F showing reduced DNA and RNA binding upon C-terminal tail removal. Arrows and cartoons indicate enzyme-substrate bands. Enzyme concentrations are indicated above each lane. Each gel-based assay in **c**- was repeated 2-3 times independently with similar results. **g** Schematic illustration proposing how the zinc-finger and the RG/RP-rich fragments within the C-terminal tail stabilize the binding of TOP3B to single-stranded DNA/RNA. **h** Experimental setup and procedure for the magnetic-tweezers experiments. -SC, negatively supercoiled DNA. F, magnetic force. Extending force used in the assay: 0.2 pN. **i** Typical time trace and abstracted contour line showing stepwise removal of DNA negative supercoils within a single TOP3B relaxation burst. **j** Histogram showing the distribution of TOP3B step-amplitude (change of DNA linking number in each step/catalytic cycle). Fitting to a Gaussian gives a mean of $0.99 \pm 0.03$ (SEM; $n = 59$). **k** Representative time-traces of a typical TOP3B multi-cycle burst and a single-cycle burst of TOP3Bcore. **l** Distributions of TOP3B and TOP3Bcore catalytic processivity (total number of DNA supercoils removed in each relaxation burst) showing the multi-cycle DNA relaxation profile of TOP3B and the single-cycle profile of TOP3Bcore.

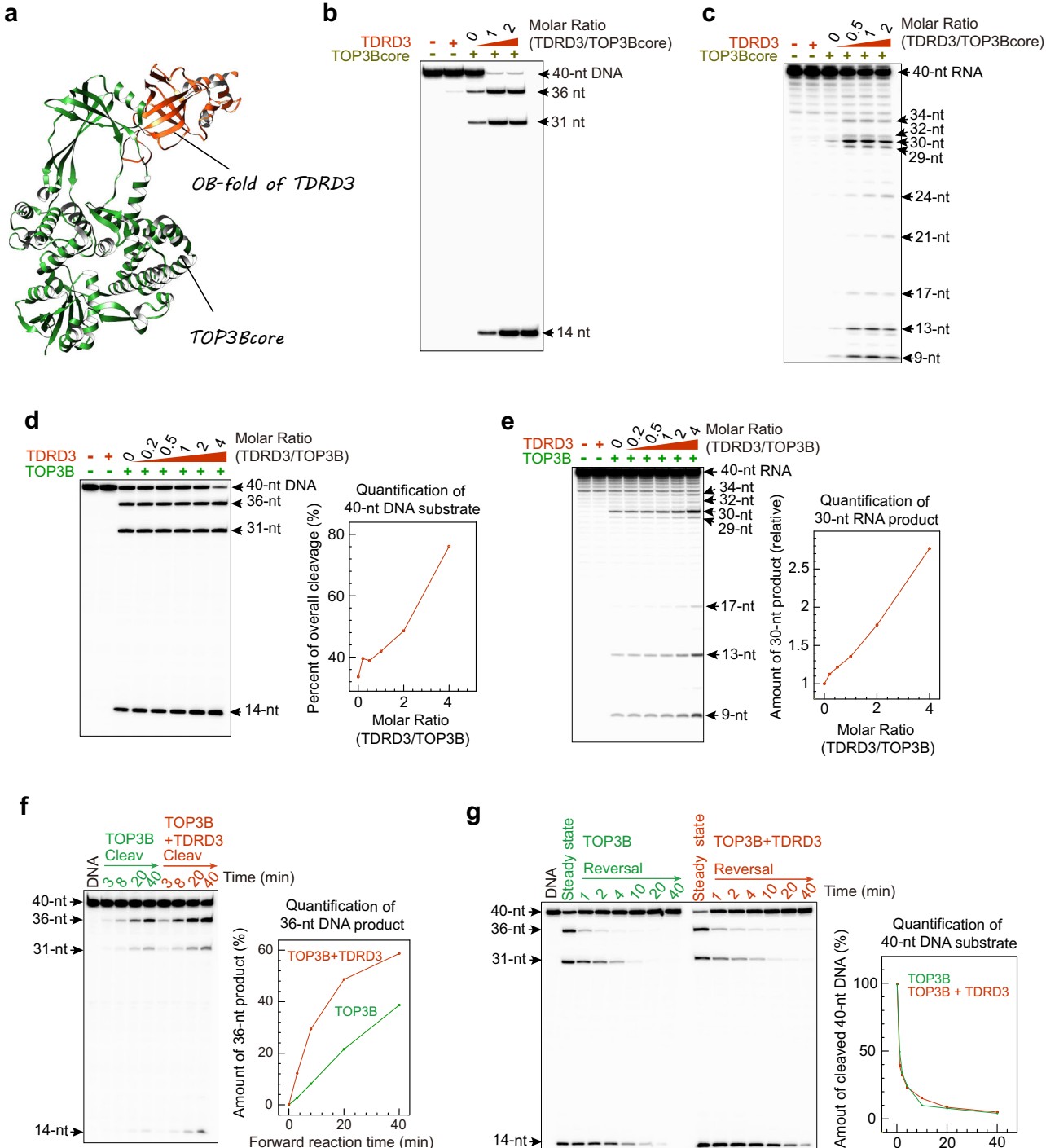

**Fig. 5 | TDRD3 enhances TOP3B-mediated DNA/RNA cleavage and has little effect on the reversal of TOP3B cleavage complexes. a** Crystal structure of TOP3Bcore in complex with the OB-fold of TDRD3 (PDB ID: 5GVE). Domain II of TOP3B forms a stable interaction with an insertion loop inside the OB-fold of TDRD3. **b, c** TDRD3 enhances TOP3Bcore-mediated DNA and RNA cleavage. Numbers above lanes indicate the TDRD3/TOP3B molar ratios. **d** Enhanced TOP3B DNA cleavage at steady state by TDRD3. Numbers indicate molar ratios of TDRD3/ TOP3B. TDRD3 titration curve (based on quantification of DNA substrate bands) is shown on the right. **e** Stimulation TOP3B-mediated RNA cleavage by TDRD3. We quantified the representative 30-nt RNA product band on this gel instead of the RNA substrate band due to the low percentage of overall RNA cleavage, to reduce

the measurement error. Y-axis values indicate the amount of 30-nt RNA product relative to the TDRD3-free lane. **f** Time-course showing the stimulation of TOP3B's forward reaction (DNA cleavage/binding) by TDRD3. The assay was conducted without added divalent metal ions. We examined the TOP3B-mediated DNA cleavage activities by tracking the formation of 36-nt DNA product (displayed as a percentage of DNA substrate). Each DNA/RNA cleavage assay in (**b–f**) were repeated twice independently with similar results. **g** Reversal of TOP3B cleavage products in the absence and presence of TDRD3 (TOP3B:TDRD3 = 1:4). Reversal plots were obtained by quantifying the 40-nt DNA substrates at different time points relative to the initial cleavage products (percentage) over time. Each dot represents a mean value from two experimental replicates.

RNA cleavage by full-length TOP3B were also markedly enhanced upon TDRD3 addition (Fig. 5d, e). We then measured the forward (cleavage) reaction rate by conducting the TOP3B cleavage assays in a divalent cation-free solution to prevent DNA rejoining. Cleavage was accelerated by TDRD3 (Fig. 5f), indicating that the enhancement of steady-state DNA cleavage by TDRD3 results at least in part from an enhancement of the forward (cleavage) reaction of TOP3B. To check the reversal rate, we conducted DNA cleavage and high-salt reversal assays with and without TDRD3 (Fig. 5g). Curves showing reversal of cleaved DNA by TOP3B over time under both conditions indicate that TDRD3 has no significant effect on the TOP3B-mediated DNA rejoining.

We next conducted both ensemble and single-molecule assays to analyze the stimulation of TOP3B supercoil relaxation activity by TDRD3. We first compared DNA relaxation by TOP3B/TOP3Bcore with and without TDRD3 addition in a gel-based assay (Fig. 6a, b). Human TOP1[53] was used as positive control to identify fully relaxed DNA on the gel. We found that TOP3B alone removes a limited number of DNA supercoils. It stalls at a particular degree of DNA superhelical density, despite excess enzyme concentration. This limited activity is likely due to the inadequate availability of ssDNA when a plasmid becomes less underwound[54]. TDRD3 distinctly improved the extent of TOP3B negative supercoil relaxation, presumably by stabilizing the TOP3B-ssDNA complex. Similarly, the extent of DNA relaxation by TOP3Bcore was also significantly enhanced by TDRD3, and the TOP3Bcore-TDRD3 even appeared to relax negatively supercoiled DNA to a greater extent than TOP3B-TDRD3.

Likewise, in the single-molecule assay (Fig. 4), TOP3B alone partially relaxed negatively supercoiled DNA (containing a mismatch bubble) in a single DNA binding and relaxation event (Figs. 4 and 6c). Addition of TDRD3 significantly improved the catalytic processivity, allowing a single TOP3B enzyme to bind and relax supercoiled DNA fully or practically fully in a single processive burst (Fig. 6d). Additionally, the TOP3B-TDRD3 complex appeared to remain bound to DNA after a full relaxation event, as re-introducing DNA supercoils (via magnetic-bead rotation) on an TOP3B-TDRD3 relaxed DNA tether resulted in an immediate stepwise relaxation of the introduced supercoils by the tightly bound TOP3B-TDRD3 complex (Fig. 6d). Similarly, TDRD3 fully activated the TOP3core enzyme, resulting in processive supercoil relaxation comparable to the WT enzyme (Fig. 6e). DNA relaxation cycling time analysis indicated that TDRD3 significantly increases TOP3B's DNA relaxation rate (Fig. 6f, g). This accelerated DNA relaxation may be achieved via structural modifications of TOP3B imposed by TDRD3 binding. Indeed, as mentioned above (see Fig. 5a), TDRD3 interacts with TOP3B[11], presumably at the hinge of the Type IA topoisomerase gate[1,26,55]. Thus, TDRD3 may reduce the flexibility of the TOP3B gate during the DNA relaxation process such as to limit the gate opening lifetime after DNA cleavage to facilitate the capture of a passing DNA strand in the TOP3B central hole and/or to promote the rejoining of the cleaved DNA strand.

Interestingly, the TOP3Bcore-TDRD3 combination consistently outperformed the TOP3B-TDRD3 combination, in both the extent and rate of negative supercoil relaxation in ensemble and single-molecule measurements, respectively (Fig. 6a, g, h). This suggests that the TOP3Bcore may form a more stable complex with TDRD3 than the full-length TOP3B, and that the C-terminal tail of TOP3B may partially counteract the binding and catalytic stimulation by TDRD3.

Together these experiments show that TDRD3 binding stabilizes the TOP3B-substrate complex and enhances the catalytic activity of TOP3B. These effects could result from two complementary and non-exclusive mechanisms: 1) increase of the DNA/RNA gate flexibility and opening-closing dynamics of TOP3B, multiplying the catalytic rate of TOP3B; 2) as TDRD3 binds both TOP3B and single-strand DNA/RNA[40,56], it could serve as a bridge between TOP3B and the nucleic acid

substrate, stabilizing the TOP3B-substrate complex for cleavage (Fig. 6j, k).

**Structure and base-sequence preferences of TOP3B for DNA/RNA substrates affect TOP3B binding and catalysis.** Although topoisomerases need to act on a variety of nucleic acid substrates, the topoisomerases that have been analyzed (including Type I and Type II topoisomerases) all show bias for preferred substrates and base sequences around their cleavage sites[5,57–59]. Type IA topoisomerase enzymes also prefer some DNA base combinations for cleavage and have been considered more active at those DNA sites[19,31,59,60]. As prokaryotic type IA topoisomerase accommodates a ~8-nt ssDNA fragment and interacts with both the DNA phosphate backbone and bases[9,12,24] in its DNA binding groove, we hypothesized that base composition would affect TOP3B enzyme-substrate pairing.

To determine the DNA-base selectivity of TOP3B, we collected 15 strong cleavage sites of TOP3B mapped in our DNA cleavage assays and aligned them from the −8 to +3 base positions (Fig. 7a). Preferential bases were observed for pyrimidines at several positions: primarily C-5, T + 2, T + 3 and T-2 (Fig. 7c). The determining role of these bases was validated by single-base replacements within a TOP3B recognition site bearing the consensus sequence, with C-5 replacement suppressing DNA cleavage (Suppl Fig. 8). Reduction of DNA cleavage was largely due to reduced DNA binding affinity, based on results from EMSA assays with the TOP3B-Y336F mutant (Fig. 7e).

To determine the RNA cleavage selectivity, we tested seven human small RNAs as TOP3B substrates (Fig. 7b). We transcribed these small RNA molecules in vitro and conducted TOP3B cleavage assays (Suppl Fig. 9). Thirty cleavage sites were collected and assigned using the RNA mapping method (Suppl Fig. 9b). Aligning the twenty-two most intense RNA cleavage sites revealed a significantly different base selectivity profile compared to DNA (compare Fig. 7d and c), where adenine bases appear frequently. Differences in TOP3B base selectivity between DNA and RNA likely relate to the different backbone compositions and conformations of RNA vs. DNA, leading to distinct bonding features upon substrate association. Indeed, simple thymine→uracil base replacements within a DNA sequence did not alter the DNA cleavage profile to RNA cleavage profile (Suppl Fig. 10).

We also compared the DNA cleavage selectivity of TOP3B with TOP3A, EcTOP1 and EcTOP3 (Fig. 7f) and found that TOP3A and TOP3B are the most similar in DNA sequence recognition, whereas EcTOP1 and EcTOP3 produced different cleavage products from TOP3B and TOP3A, as well as from each other. These results likely reflect the comparable residue compositions inside the DNA binding grooves of TOP3A and TOP3B.

The four Type IA enzymes showed marked differences with respect to RNA cleavage (Fig. 7g). TOP3A failed to generate detectable RNA cleavage products, despite being active under parallel conditions in the DNA cleavage assays (Fig. 7f). EcTOP1 and EcTOP3 also behave differently, with only EcTOP3 cleaving RNA (Fig. 7g). These results further establish that TOP3B but not TOP3A acts as human RNA topoisomerase.

We also tested how the secondary structure of a nucleic acid substrate affects TOP3B cleavage (Suppl Fig. 11). We designed various DNA substrates including particular secondary structure elements, such as a DNA stem-loop, DNA bulge or a DNA mismatch bubble, together with a TOP3B recognition sequence to obtain site-specific cleavage (Suppl Fig. 11a). We found that TOP3B cleaved at the designed cleavage site only when the latter was single-stranded or partially single stranded. Annealing of a TOP3B recognition site to a complementary strand efficiently suppressed TOP3B cleavage. These results demonstrate that TOP3B, like other type IA enzymes, requires its substrate to be single stranded to occupy the active site of the enzyme and be processed (Suppl Fig. 11a, b).

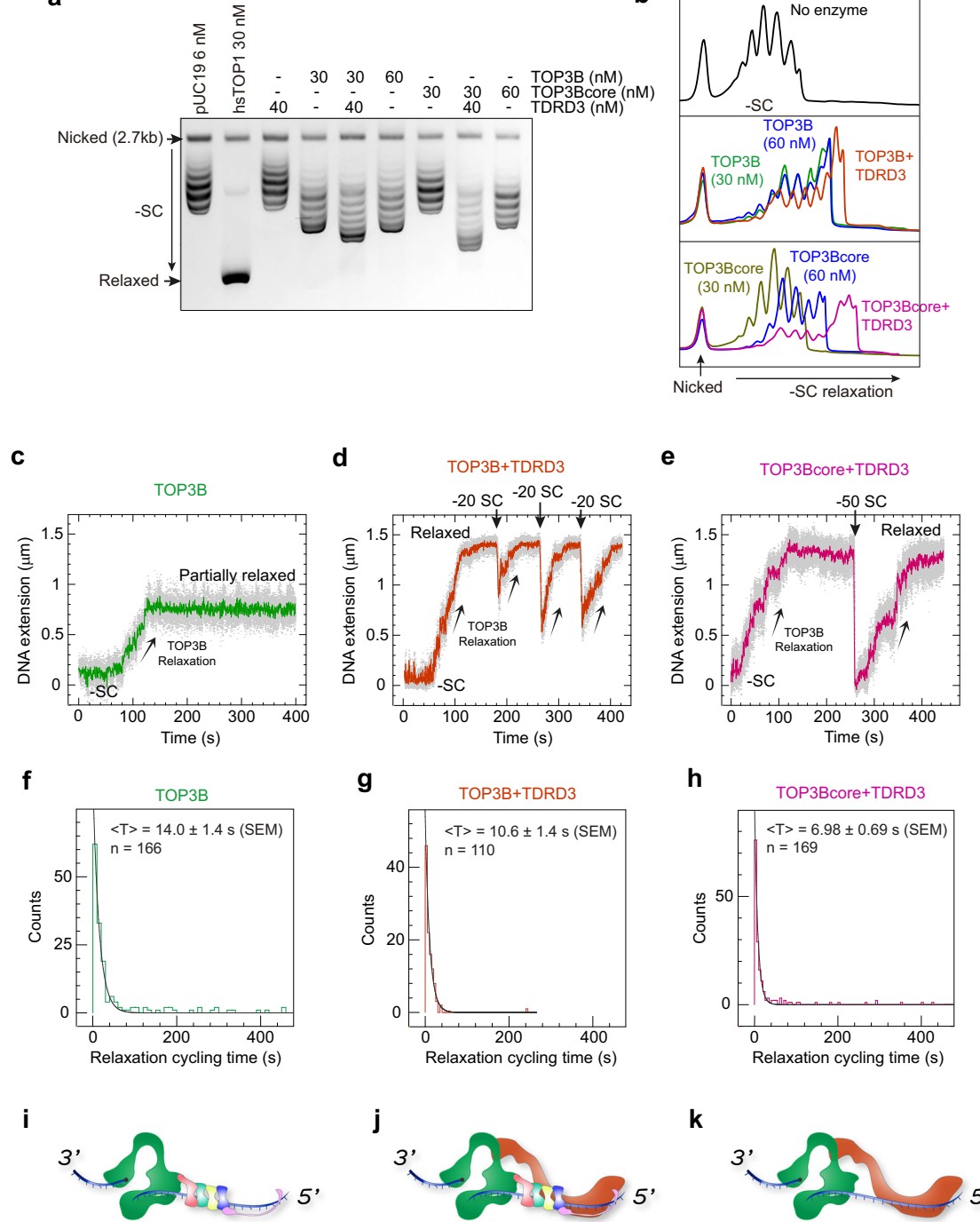

**Fig. 6 | TDRD3 enhances both the processivity and catalytic rate of TOP3B.**
**a** Gel-based plasmid relaxation assay with TOP3B and TOP3Bcore in the presence and absence of TDRD3 (repeated twice independently with similar results). Ethidium bromide (0.15 µg/ml) was added to both the agarose gel and running buffer to convert the negatively supercoiled (-SC) DNA substrate and products into slightly overwound topoisomers. Fully relaxed plasmids by hsTOP1 were included (second lane) as a positive control. Numbers indicate the concentrations of DNA substrate and enzymes used. DNA relaxation assays were performed at 37 °C for 1 hr.
**b** Quantification of DNA topoisomer distributions. Lane profiles were obtained with ImageJ software. Peaks represent individual topoisomers. Overlapping product peaks allow direct observation of changes of DNA supercoiling levels catalyzed by the indicated enzyme combinations. **c**–**e** Representative single-molecule DNA relaxation time-traces showing partial DNA relaxation by TOP3B alone and full relaxation by TOP3B and TOP3Bcore in the presence of TDRD3. Upward arrows indicate TOP3B mediated DNA relaxation and down arrows the re-introduction of DNA negative supercoils via rotation of the magnets. Numbers in (**d**, **e**) indicate the amount of supercoils introduced at each timepoint. **f**–**h** Histograms showing distributions of enzyme catalytic-cycle times with the indicated enzyme combinations. Black solid curves indicate exponential decay fits yielding a mean of 14.0 ± 1.4 s (SEM; $n = 166$) for TOP3B alone, 10.6 ± 1.4 s (SEM; $n = 110$) for TOP3B + TDRD3, and 6.98 ± 0.69 s (SEM; $n = 169$) for TOP3Bcore + TDRD3. **i**–**k** Schematic models of enzyme-DNA assembly with TOP3B alone, TOP3B + TDRD3 and TOP3Bcore + TDRD3.

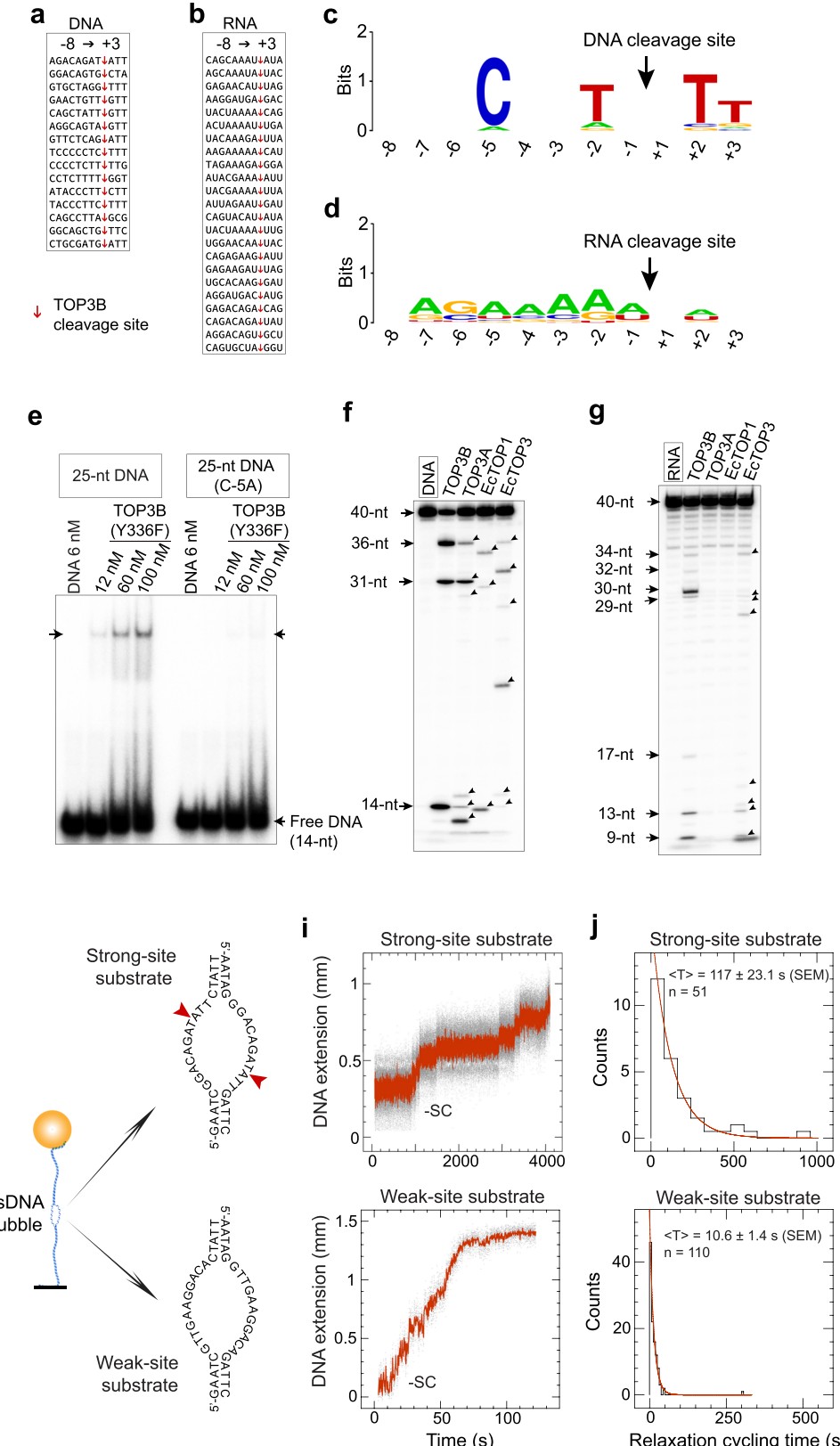

Our next question was whether strong cleavage sites correspond to effective catalytic sites of TOP3B. We thus conducted single-molecule DNA relaxation assays (see Fig. 4h) to directly measure the catalytic-cycling parameters of TOP3B using two DNA mismatch bubbles: one containing 2 strong TOP3B sites and the other containing weak sites (Fig. 7h). After verifying TOP3B-mediated cleavage profiles in oligonucleotide assays (Suppl Fig. 12), we inserted these bubble segments into two 6-kb DNA tethers and carried out DNA relaxation assays with TOP3B in the presence of TDRD3 to stimulate the catalytic activity of TOP3B. Remarkably, we observed that relaxation cycling times of TOP3B with the strong-site substrate are much longer than those with

**Fig. 7 | Nucleic acid sequence influences TOP3B binding, cleavage and catalytic cycling rate. a**, Alignment (−8→+3 base positions) of 15 strong DNA cleavage sites of TOP3B (determined with sequence-specific DNA oligonucleotides as markers). Cleavage sites are indicated by red arrows. **b** Alignment of TOP3B RNA cleavage sites. Cleavage sites (arrows) were mapped with RNA alkaline hydrolysis ladders and RNase T1 ladders (see Suppl Figs. 2 and 9). **c, d** DNA and RNA sequence logos showing TOP3B base selectivity were generated with WebLogo online server, based on the sequences shown in panels **a** and **b**. DNA/RNA base positions and TOP3B cleavage sites are indicated by numbers and arrows. **e** TOP3B (Y336F mutant) has a higher affinity for an oligonucleotide with a strong than a weak TOP3B cleavage site. The weak-site oligonucleotide was generated by replacing a C with an A base within a strong-site containing oligonucleotide (see Suppl Fig. 8). **f, g** DNA/RNA cleavage profiles of TOP3B, TOP3A, EcTOP1 and EcTOP3. Numbers indicate the mapped DNA and RNA cleavage sites of TOP3B with a 40-nt DNA substrate and the RNA counterpart. Arrowheads indicate the cleavage products of hsTOP3A, EcTOP1 and EcTOP3. Each gel-based assay in **e–g** were repeated twice independently with similar results. **h** Design of ssDNA mismatch bubble substrates for single-molecule experiments. A dsDNA tether made with a DNA mismatch bubble containing either strong (top) or weak (bottom) TOP3B recognition sites. **i, j** Single-molecule TOP3B relaxation time-traces and catalytic-cycle time distributions collected with the strong-site and weak-site containing DNA tethers. Fitting to an exponential decay yielded in a mean of $10.6 \pm 1.4$ s (SEM; $n = 110$) for the weak-site substrate and $117.0 \pm 23.1$ s (SEM; $n = 51$) for the strong-site substrate.

the weak-site substrate (Fig. 7i). Fitting exponential decays to their turnover-lifetime histograms (Fig. 7j) gave an average lifetime value of $117 \pm 23.1$ s (SEM; $n = 51$) for the strong-site turnovers, ten-fold longer than for the weak-site turnovers ($10.6 \pm 1.4$ s, SEM; $n = 110$). This suggests that a strong recognition site negatively affects TOP3's catalytic efficiency. When combining the following three observations showing that TOP3B has (1) a slower catalytic cycling rate when bound to a strong recognition site, (2) elevated amount of steady-state cleavage products at a strong site, (3) a relatively long DNA reversal time when cleaves at a strong recognition site (Fig. 5g), we postulate that a strong site slows down DNA end-rejoining by TOP3B and temporarily traps the enzyme at the post-cleavage state. The slow rejoining at a strong recognition site is likely due to an excessive constraint of the cleaved ssDNA strands by the enzyme, reducing their plasticity and preventing efficient DNA end alignment for rejoining.

## Discussion

In this study, we investigated basic catalytic mechanisms of TOP3B, including the roles of divalent metal ions and of the conserved K10 residue. We also examined how the basic DNA and RNA topoisomerase activities of TOP3B are modulated by its C-terminal domains, its protein partner TDRD3 and the sequence and structure of its nucleic acid substrates.

Although a divalent cation is commonly seen in the catalytic center of type IA topoisomerases, there is currently no consensus on the absolute requirement of metal ions for DNA cleavage[12,14,18,19]. Although we and others observed that Type IA topoisomerase enzymes can cleave DNA without adding divalent ions in solution, minimal amounts of divalent cations in the reactions (derived from purified enzymes, substrates or reaction buffer) may effectively trigger DNA cleavage due to the presumably high affinity for the enzymes for the metals in their catalytic site. Accordingly, we observed that addition of $Mg^{2+}$ and to a greater extent $Mn^{2+}$ stimulated DNA binding and cleavage by TOP3B (Figs. 1–3), and that excess EDTA efficiently suppressed DNA and RNA cleavage (Suppl Fig. 3). In addition, our results with the mutant TOP3B-K10M and the mutant TOP3B-E9Q provide direct evidence for the catalytic role of divalent cations in TOP3B's DNA cleavage activity (Fig. 2 and Suppl. Figure 4). We thus suggest that divalent ions play crucial roles in triggering TOP3B cleavage.

Although we proposed a single metal ion mechanism for TOP3B catalysis, which is based on the resolved Type IA topoisomerase structures, it is plausible that a second or more metal ions are involved, especially in the DNA end-rejoining process, considering that we observed that DNA rejoining by TOP3B is triggered by a moderate $Mg^{2+}$ addition and the metal remained unsaturated at a relatively high concentration (Fig. 1f, g). Therefore, it is likely that TOP3B requires multiple divalent cations for nucleic acid rejoining. $Mn^{2+}$ may be more effective in occupying multiple metal binding sites, which would account for the fact that it can fully activate TOP3B reversal at a lower concentration (Fig. 1f, g). In addition, RNA but not DNA can be rejoined

by TOP3B with minimal amount of divalent cations in the reaction system (Fig. 1h), implying that RNA may require fewer metal ions for catalysis owing to its relatively high chemical reactivity than DNA. It is worth noting that a two-metal binding state of type II topoisomerases was recently reported[14] which may help predicting a possible multiple-metal strategy for Type IA topoisomerases. On the other hand, a possible reason that one cannot easily observe additional metal ions bound to the enzyme is their short lifetime of association. For example, nucleases previously believed to adopt classical two-metal catalysis such as RNase H1 were recently found to require transient associations of additional metal ions to be fully active[16]. Capture of these trafficking metal cations requires approaches with high temporal resolution.

Our study on the TOP3B-K10M mutant revealed the differential impact of the K10 catalytic residue on topoisomerase activities of TOP3B on DNA and RNA. In the presence of divalent ions, TOP3B-K10M could cleave both DNA and RNA while only RNA could be rejoined. Thus, the TOP3B-K10M mutant may be exploited as a tool to differentially study the DNA and RNA related activities of TOP3B in cells, and as a tool for structural studies of TOP3B cleavage complexes.

An unexpected observation in our study is that the TOP3B-core + TDRD3 combination consistently shows better catalytic behaviors than the TOP3B + TDRD3 combination in both DNA/RNA cleavage (Figs. 5b-c vs. 5d-e) and DNA relaxation (Fig. 6a, g, h). This implies that the TOP3Bcore may form a more stable complex with TDRD3 than the full-length TOP3B. A given amount of TDRD3 may sufficiently saturate the TOP3Bcore to achieve maximum DNA/RNA cleavage (Fig. 5b, c), while the full-length TOP3B may require excess TDRD3 (Fig. 5d, e). Therefore, the C-terminal domains of TOP3B may negatively affect the stability of TOP3B-TDRD3-DNA/RNA complex (in addition to post translational modifications[61]). This suggests that the TOP3B-TDRD3 interaction may be dynamically regulated in cells. TDRD3-free TOP3B molecules may act alone or directly bind and collaborate with other protein partners like RNA/DNA helicases.

## Methods
### Cloning, protein expression and purification
Protein expression vectors for hsTOP3B, TOP3B-K10M, TOP3B-E9Q, TOP3B(1-612AA), TOP3B(1-823AA), TOP3B-Y336F, TOP3B-Y336F(1-612AA), hsTOP3A, and hsTDRD3, were cloned into a modified pLEXm plasmid, including an N-terminal His$_8$-MBP-tag[15]. Vector containing recombinant DNA was transfected into human HEK293T cells for protein expression. Detailed methods for vector transfection, cell collection and lysis were described previously[15]. Lysis buffer contains 20 mM Tris-HCl (pH 7.8), 0.5 M KCl, 0.1 mM EDTA, 0.05% Tween 20, 2.5% Glycerol, 2 mM DTT, 1 mM PMSF, and 1 X protease inhibitor cocktail (Roche). Clarified supernatant of the lysate was then incubated with 7.5 ml amylose resin (NEB), washed with 400 ml wash buffer (Lysis buffer lacking EDTA and protease inhibitors) and finally eluted with Elution buffer (wash buffer with 40 mM maltose). Eluted protein sample was then concentrated to 1 ml volume via ultrafiltration and

run over a Superdex 200 gel filtration column (GE Healthcare) in 20 mM Tris-HCl (pH 7.8), 0.4 M KCl, 0.05% Tween 20, 5% Glycerol, and 2 mM DTT, to remove protein aggregates. Glycerol in the purified protein samples were adjusted to 25% before liquid-nitrogen snap freezing and storing at −80 °C.

## DNA/RNA oligo labeling and TOP3B cleavage assays

DNA/RNA oligos were labeled with $^{32}$P at the 5′ end using ATP-γ–$^{32}$P (PerkinElmer) and T4 PNK kinase (NEB) according to the NEB online protocol for radioactive labeling. Oligos after labeling were purified using a mini Quick Spin DNA Column (Roche) to remove unincorporated ATP-γ-$^{32}$P. TOP3B DNA cleavage assay was carried out in a 20 µl reaction containing 5 mM Tris-HCl (pH 7.5), 100 mM potassium glutamate, 0.02% Tween 20, 2 mM DTT, with addition of indicated amount of MnCl$_2$/MgCl$_2$/EDTA. Buffer for RNA cleavage assays contains 1X RNasin® Plus Ribonuclease Inhibitor (Promega). Final concentrations of labeled DNA/RNA oligo and TOP3B/TOP3B mutant were 6 nM and 18 nM. After incubation at 30 °C for 30 min or other indicated period of time, cleavage products (20 µl) were mixed with 20 µl 2X Formamide gel-loading buffer (10 mM EDTA, 0.025% bromophenol blue, 0.025% Xylene cyanol FF and 0.2% SDS dissolved in formamide), heat denatured at 95 °C for 3 min, and separated on a 18% acrylamide gel containing 7 M Urea. Gel was then dried and imaged with a GE Typhoon Phosphorimager.

To observe reversal of TOP3B cleavage product, 2.2 µl of 10X reversal buffer containing 5 mM Tris-HCl (pH 7.5), 5 M NaCl, and indicated metal/EDTA concentration was added to the 20 µl reaction sample and incubated for required time before being terminated by mixing with the 2X Formamide gel-loading buffer.

## Small RNAs in vitro transcription

DNA oligonucleotides containing small RNA sequences and a T7 promoter fragment at the 5′ end were annealed with the complimentary DNA strand to produce dsDNA templates for in vitro transcription using the HiScribe T7 Quick High Yield RNA Synthesis Kit (NEB). Small RNA transcripts were generated as described by the manufacturer and purified with Monarch® RNA cleanup kit (NEB). The purified RNA were then treated with shrimp alkaline phosphatase (NEB) to remove the 5′-triphosphate groups, and labeled with $^{32}$P at the 5′ end using ATP-γ-32P (PerkinElmer) and T4 PNK kinase (NEB) as described above.

## DNA and RNA Electrophoretic mobility shift assay

6 nM $^{32}$P-labeled DNA/RNA oligo was incubated with various amount of TOP3B/TOP3Bcore, in 10 mM Tris-HCl (pH 8.0), 40 mM KCl, 0.01% Tween 20, 6% Glycerol, 0.01% BSA, 2 mM DTT, and requested concentration of divalent cations/EDTA, at 30 °C for 30 min. Buffer for RNA assay contains 1X RNasin® Plus Ribonuclease Inhibitor (Promega). Reaction sample was placed on ice for 5 min before loading on a pre-run 6% DNA retardation gel (Invitrogen) at 4 °C. Electrophoresis was conducted at 200 V for 45 min in the gel running buffer containing 0.5X TBE with MnCl$_2$ addition (Mn$^{2+}$ concentration was adjusted to 0.1 mM).

## Gel-based TOP3B relaxation assay

6 nM negatively supercoiled pUC19 plasmid was incubated with various amounts of TOP3B, TOP3Bcore, TDRD3 or their combinations, at 37 °C for 1 hr, in 10 mM Hepes pH 7.5, 20 mM potassium glutamate, 0.02% Tween 20, 0.01% BSA, and 2 mM DTT (20 µl reaction volume). Reactions were stopped via introduction of 0.4 M KCl for 5 min and followed by additions of SDS (to 0.1%) and 6X DNA loading buffer (NEB). Aliquots of Reaction products were loaded on a 1% agarose gel and run at 60 V for 300 min. Both the gel and running buffer (1X TBE) contain 0.15 µg/ml EtBr. hsTOP1 relaxation assay with the same plasmid and buffer condition was conducted as positive control.

## Single-molecule TOP3B DNA relaxation assays

A 6-kb bubble-containing DNA tether for magnetic tweezers assay was obtained by ligating two 3-kb dsDNA handles to the two ends of a short DNA insert containing a 11-base mismatch bubble. The 3-kb DNA handles were parts of pET28b plasmid and obtained via PCR. The 6-kb DNA tether was then ligated to a multi-biotin labeled 500-bp DNA fragment at one end and multi-digoxigenin labeled 500-bp DNA at the other. DNA sticky ends for ligation were generated by BsaI-HF (NEB) digestions. The biotin labeled DNA fragment was then attached to a 1 µm streptavidin coated magnetic beads (MyOne T1, Invitrogen) and the digoxigenin labeled end was attached to an anti-dig covered bottom glass surface of a sample flow cell, as described previously[26]. A pair of permanent magnets sit above the flow cell provides magnetic force on the magnetic bead to extend the DNA tether in solution. DNA supercoiling was introduced by horizontally rotating the magnets that leads to formation of DNA plectonemes and reduction of the DNA extension. The real-time alteration of DNA extension was recorded by monitoring the size of the diffraction ring of the magnetic bead via an inverted microscope that is connected to a CCD camera[51]. TOP3B relaxation assays were conducted at 35 °C, in 10 mM Hepes pH 7.5, 90 mM potassium glutamate, 0.02% Tween 20, 0.01% BSA, and 2 mM DTT. Data collection and processing was described previously[52] using the Xvin software suite (PicoTwist SARL).

## RADAR (rapid approach to DNA adduct recovery) assay

RADAR assay was performed for detection of TOP3Bccs as described previously. HEK293 cells (1 × 10$^6$) transfected with TOP3B/TOP3B-K10M/TOP3B-R338W expression vector for 72 hrs were lysed with 1 mL DNAzol (ThermoFisher Scientific). Nucleic acids were precipitated via centrifugation (12,000 × g for 10 min) following the addition of 0.5 mL of 100% ethanol. Precipitates were washed twice with 75% ethanol, resuspended in 200 µl TE buffer, heated at 65 °C for 15 minutes. After sonication (40% power for 15 s pulse and 30 s rest 5 times) and centrifugation (15,000 rpm for 5 min), supernatant containing nucleic acids and covalently linked proteins was collected, quantitated and slot blotted. TOP3B cleavage complexes were detected with diluted (1:1000) rabbit monoclonal Anti-TOP3B antibody [EP7779] (Abcam).

## Reporting summary

Further information on research design is available in the Nature Research Reporting Summary linked to this article.

## Data availability

The data that support the findings of this study are available in the manuscript and its supplementary information. The structural data for the TOP3B-TDRD3 protein complex used in Fig. 5 is available in the Protein Data Bank under the accession code 5GVE. All data are available from the authors upon reasonable request. Source data are provided with this paper.

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

## Acknowledgements
We are grateful to Xuemin Chen (NIDDK, NIH) for his assistance on human cell culture and protein purification, and to Yeonee Seol (NHLBI, NIH) for her help on the single-molecule experimental set up and sample flow cell assembly. We thank Yuk-Ching Tse-Dinh (Florida International University) for EcTOP1 enzyme and Yilun Sun (NCI, NIH) for technical assistance in the experiments with hsTOP1 enzyme. We thank Keli Agama, Shar-Yin Huang, and Hongliang Zhang in our laboratory for sharing and discussing preliminary work on TOP3B with DNA substrates for the TOP3B cleavage assays. Our work is supported by the Center for Cancer Research (CCR), the Intramural Program of the National Cancer Institute (Z01-BC-006161) to Y.P.

## Author contributions
X.Y., W.Y., K.C.N., and Y.P. designed the biochemical and single-molecule assays. X.Y. performed the biochemical and single-molecule experiments. S.S. performed the cell-based RADAR assay. X.Y., K.C.N., and Y.P. analyzed data from the gel-based DNA/RNA cleavage assays and the single-molecule assays. X.Y., W.Y., K.C.N., and Y.P. wrote the manuscript.

## Funding

## Competing interests
The authors declare no competing interests.
