## [Peer Review File · Nature Communications]

Structural and biochemical basis for DNA and RNA catalysis by human Topoisomerase 3 βREVIEWER COMMENTS

Reviewer #1 (Remarks to the Author):

The manuscript by Yang et al. describes the biochemical mechanism of Human TOP3B, an important DNA topoisomerase for several cellular processes including meiosis and DNA repair, that has been shown to act also on RNA.

This study contains a substantial and impressive amount of biochemical data combined with single-molecule experiments that allow to clarify how TOP3B is able to perform its specific catalytic activities, a question that was raised in a 2017 study by Goto-Ito et al. (reference 11). The data presented in this manuscript have been generated with a rigorous and systematic experimental design. The authors analyzed thoroughly the role of metal ions and the role of a conserved lysine in the catalysis of DNA/RNA substrates, and the structural modulation of the catalytic reaction, involving interaction with its auxiliary factor TDRD3.

You will find below some minor comments or questions:

- line 148-149: the authors mention that the 2'-OH groups make RNA more rigid than in the case of DNA. Since both DNA and RNA would be anyway embedded in the TOP3B nucleic acid groove, is there any residue that would further stabilize the 2'-OH groups and provide additional interactions apart from the catalytic residues ?

- figure 5: the authors are presenting a quantification analysis of the cleavage bands. Could the authors explain the choice of the cleavage band for quantification since they are different in each panel d-g ?

- figure 6: the authors compared DNA relaxation using single-molecule assays of TOP3B alone and TOP3B full length / core with TDRD3. It is clear that TDRD3 stimulates relaxation by the level of DNA extension observed between panels c and d/e in the first 100-200 sec of the analysis. However panel c, d, e have different conditions in term of re-introduction of supercoils (0, -20 and -50), could the authors explain the reason for probing these different parameters, and if they tried the same sequence for all 3, how the complexes behaved ?

- line 226-232 : the authors discuss the accelerated DNA relaxation rate in presence of TDRD3. They make the hypothesis that TDRD3 may increase the flexibility of the TOP3B DNA-gate to promote the passage of the uncleaved strand through the break. If TDRD3 sits on the hinge region of the toroidal structure of TOP3B as shown in ref 11, one would expect a limitation rather than an enhancement of the flexibility of the DNA-gate. Could it be that TDRD3 instead stabilizes a structure that promotes strand passage or rejoining of DNA ends ? In ref 11, is TDRD3 stabilizing a particular conformation of TOP3B when compared to other available structures ? Is it known if the same effect is expected for decatenation ?

- The DNA templates used for the experiments from Suppl figure 11 are very interesting since they are closer to physiological topologies that may occur in the cell, when most studies of DNA cleavage are done on single-stranded oligonucleotides. The results from this figure should be more detailed in the main text.

The authors have used the full length TOP3B to analyze the cleavage pattern with these scaffolds. This might be beyond the scope of this study, but would the authors expect a similar pattern with the TOP3Bcore or TOP3B-TDRD3 complex and do they expect that they would fit with the schematic models of Figure 6i,j,k ?

Reviewer #2 (Remarks to the Author):

The manuscript of Yang et al. presents a biochemical and single molecule analysis of human topoisomerase 3beta. The interest in this enzyme stems from the fact that it is an RNA topoisomerase. Topo3 is a member of the type IA family of topoisomerases, which have been characterized extensively using a variety of approaches. The work presents information on unique aspects of human topo3, such as metal requirements and the role of the C-terminal domain. The combination of

biochemical experiments and single molecule measurements is very strong and provides new insights on this enzyme. Nonetheless, there are no major new insights from the experiments. For example, the role of metals in cleavage of DNA/RNA is still as ambiguous as it was before. Overall, this is excellent work that helps understand an important enzyme in much more detail but the nature of the experiments and the overall conclusions make the manuscript more appropriate for a more specialized journal.

REVIEWER COMMENTS

Reviewer #1 (Remarks to the Author):

The manuscript by Yang et al. describes the biochemical mechanism of Human TOP3B, an important DNA topoisomerase for several cellular processes including meiosis and DNA repair, that has been shown to act also on RNA.

This study contains a substantial and impressive amount of biochemical data combined with single-molecule experiments that allow to clarify how TOP3B is able to perform its specific catalytic activities, a question that was raised in a 2017 study by Goto-Ito et al. (reference 11). The data presented in this manuscript have been generated with a rigorous and systematic experimental design. The authors analyzed thoroughly the role of metal ions and the role of a conserved lysine in the catalysis of DNA/RNA substrates, and the structural modulation of the catalytic reaction, involving interaction with its auxiliary factor TDRD3.

Our answer: Thank you for finding our study substantial, and for your supportive comments.

You will find below some minor comments or questions:

- line 148-149: the authors mention that the 2'-OH groups make RNA more rigid than in the case of DNA. Since both DNA and RNA would be anyway embedded in the TOP3B nucleic acid groove, is there any residue that would further stabilize the 2'-OH groups and provide additional interactions apart from the catalytic residues?

Our answer: Excellent question! It is highly possible that the 2'-OH groups form additional molecular interactions with the surrounding AA residues in the TOP3B nucleic-acid binding groove. However, they may not be easily predicted using the available structural information from the Top1A-DNA complexes, as RNA and DNA have major differences concerning their backbone conformations; their association with the nucleic-acid binding groove of TOP3B may also induce slightly different local domain movements. We added this discussion in the main text (Lines 149-150).

- figure 5: the authors are presenting a quantification analysis of the cleavage bands. Could the authors explain the choice of the cleavage band for quantification since they are different in each panel d-g ?

Our answer: We choose the bands directly reflecting the overall amounts of DNA/RNA cleavage (sum of all cleavage products) (panels d and g). Additionally, in situations where the percentages of cleaved substrates were low (panels e and f), to achieve more precise measurements, we selected one of the representative cleavage products for quantification, which largely reflects the amount change of the overall DNA/RNA substrates. We have edited the figure legends (Fig5, e and f) to clarify selection of bands for quantification. Thank you for the suggestion.

- figure 6: the authors compared DNA relaxation using single-molecule assays of TOP3B alone and TOP3B full length / core with TDRD3. It is clear that TDRD3 stimulates relaxation by the level of DNA extension observed between panels c and d/e in the first 100-200 sec of the analysis. However, panel c, d, e have different conditions in term of re-introduction of supercoils (0, -20 and -50), could the authors explain the reason for probing these different parameters, and if they tried the same sequence for all 3, how the complexes behaved?

Our answer: As the reviewer points out, the initial relaxation events starting on a -SC DNA (first 100-200 s of the time traces in c, d and e) showed the stimulation of TOP3B's catalytic activity by TDRD3. The purpose of introducing additional negative supercoils was to re-create negatively supercoiled DNA substrate to observe the processive supercoil relaxation by the TOP3B enzymes. Because the DNA in Fig.6c was still negatively supercoiled, there was no need to re-introduce supercoils.

We did not introduce a fixed number of DNA supercoils. During data collection, we measured simultaneously the real-time DNA extension of multiple DNA tethers. We adjusted the number of turns added to the DNA molecules considering the supercoiling states of all the DNA tethers at a given time point. For example, we have DNA tether_A at -30 SC and DNA tether_B was just relaxed by the enzyme (0 SC). Rather than adding -40 SC to all the DNA tethers, which would supercoil DNA tether_A to -70 SC, bringing the magnetic bead attached to DNA tether_A too close to the glass surface and risking getting stuck on the surface. We would instead add -20 SC to all the DNA tethers.

In our single-molecule study, we applied a constant extending force (0.2pN) on the linear DNA tethers; thus, the torsional tension applied on the DNA (which determines the DNA structure and enzyme behavior) remained constant once the number of supercoils added to DNA falls into the linear region of a DNA extension vs. supercoiling curve (plectoneme region of the hat curve) (For more information on the DNA extension vs. supercoiling curve, see ref 52 or DOI 10.1074/jbc.R109.092437). Specifically, under the constant force conditions of the magnetic tweezers, increasing the negative supercoiling increases the length of the plectoneme region, but does not otherwise alter the structure or properties of the DNA. This is in distinct contrast with supercoiling of a plasmid, for which additional negative supercoiling changes the structure and propensity to form single-stranded DNA.

Under our extending force and buffer conditions with a 6-kb DNA tether, -20 SC falls into the linear region of the DNA hat curve, -50 SC should be slightly out of the linear region (resulting in a slightly larger torque on the DNA compared to -20 SC, but not enough to measurably vary the enzyme behavior). During our data collection, we introduced various numbers of supercoils to DNA (from -15 to -50) and did not observe differences in the DNA relaxation processivity or catalytic rate. For measurements of the enzyme cycling lifetime and catalytic step amplitude, we only selected TOP3B relaxation events within the linear region of the hat curve of a particular DNA tether, as mentioned in the Methods section and ref 52.

- line 226-232 : the authors discuss the accelerated DNA relaxation rate in presence of TDRD3. They make the hypothesis that TDRD3 may increase the flexibility of the TOP3B DNA-gate to promote the passage of the uncleaved strand through the break. If TDRD3 sits on the hinge region of the toroidal structure of TOP3B as shown in ref 11, one would expect a limitation rather than an enhancement of the flexibility of the DNA-gate. Could it be that TDRD3 instead stabilizes a structure that promotes strand passage or rejoining of DNA ends?

Our answer: Thanks for the insightful suggestion, which we have included in the revised manuscript (Lines 230-233).

In ref 11, is TDRD3 stabilizing a particular conformation of TOP3B when compared to other available structures?

Our answer: In ref 11, the TDRD3 bound TOP3B core is roughly identical to the apo structure, both in the presumed gate-closed state. (No one so far has observed and confirmed a TopIA protein structure in an open gate state.) However, it is plausible that TDRD3 binding stabilizes the gate-closed state of TOP3B during DNA relaxation. This point has been added in our revised manuscript.

Is it known if the same effect is expected for decatenation?

There is so far no direct evidence showing the effect of TDRD3 on TOP3B catenation/decatenation.

- The DNA templates used for the experiments from Suppl figure 11 are very interesting since they are closer to physiological topologies that may occur in the cell, when most studies of DNA cleavage are done on single-stranded oligonucleotides. The results from this figure should be more detailed in the main text.

Our answer: Thank you. We have included further clarification of these results in the main text accordingly (Lines 283-290).

The authors have used the full length TOP3B to analyze the cleavage pattern with these scaffolds. This might be beyond the scope of this study, but would the authors expect a similar pattern with the TOP3Bcore or TOP3B-TDRD3 complex and do they expect that they would fit with the schematic models of Figure 6i,j,k?

Our answer: Enzyme association should induce rearrangements of the overall structure of the DNA substrate to form an energetically favored protein-DNA complex. For example, in the context of TOP3B binding to a DNA stem-loop: the core domain of TOP3B binding to the ssDNA region of a DNA stem-loop may cause partial unpairing of the nearby DNA stem. This unpairing may be further extended by the association of the CTD of TOP3B, which may eventually fully or mostly unwind the DNA substrate. Likewise, DNA binding by TOP3Bcore and TOP3B-TDRD3 should also induce structural transitions in both the DNA and the protein. In this case, the models should fit. Although we provide a ssDNA in our model, a certain type of DNA secondary structure adjacent to a TOP3B cleavage site may also improve the enzyme binding and cleavage which requires further investigation. This point is mentioned in our revision.

Reviewer #2 (Remarks to the Author):

The manuscript of Yang et al. presents a biochemical and single molecule analysis of human topoisomerase 3beta. The interest in this enzyme stems from the fact that it is an RNA topoisomerase. Topo3 is a member of the type IA family of topoisomerases, which have been characterized extensively using a variety of approaches. The work presents information on unique aspects of human topo3, such as metal requirements and the role of the C-terminal domain. The combination of biochemical experiments and single molecule measurements is very strong and provides new insights on this enzyme. Nonetheless, there are no major new insights from the experiments. For example, the role of metals in cleavage of DNA/RNA is still as ambiguous as it was before. Overall, this is excellent work that helps understand an important enzyme in much more detail, but the nature of the experiments and the overall conclusions make the manuscript more appropriate for a more specialized journal.

Our answer: Thank you for finding our combination of biochemical experiments and single-molecule measurements very strong and our “work excellent”, which “helps understand an important enzyme”.

REVIEWERS' COMMENTS

Reviewer #1 (Remarks to the Author):

All the comments have been appropriately addressed by the authors with additional explanations in the main text or figure legends.

This manuscript is clear and insightful.

Point-by-point response to the reviewers' comments:

Reviewer #1 (Remarks to the Author):

All the comments have been appropriately addressed by the authors with additional explanations in the main text or figure legends.

This manuscript is clear and insightful.

Our answer: Thank you for finding our manuscript clear and insightful and thank you again for your excellent questions and comments that improved the quality of our manuscript.